# Carboxymethyl Bacterial Cellulose from Nata de Coco: Effects of NaOH

**DOI:** 10.3390/polym13030348

**Published:** 2021-01-22

**Authors:** Pornchai Rachtanapun, Pensak Jantrawut, Warinporn Klunklin, Kittisak Jantanasakulwong, Yuthana Phimolsiripol, Noppol Leksawasdi, Phisit Seesuriyachan, Thanongsak Chaiyaso, Chayatip Insomphun, Suphat Phongthai, Sarana Rose Sommano, Winita Punyodom, Alissara Reungsang, Thi Minh Phuong Ngo

**Affiliations:** 1Faculty of Agro-Industry, School of Agro-Industry, Chiang Mai University, Chiang Mai 50100, Thailand; warinporn.k@cmu.ac.th (W.K.); jantanasakulwong.k@gmail.com (K.J.); yuthana.p@cmu.ac.th (Y.P.); noppol@hotmail.com (N.L.); phisit.s@cmu.ac.th (P.S.); thachaiyaso@hotmail.com (T.C.); chayatip@yahoo.com (C.I.); suphat.phongthai@cmu.ac.th (S.P.); 2The Cluster of Agro Bio-Circular-Green Industry (Agro BCG), Chiang Mai University, Chiang Mai 50100, Thailand; pensak.amuamu@gmail.com; 3Center of Excellence in Materials Science and Technology, Chiang Mai University, Chiang Mai 50200, Thailand; sarana.s@cmu.ac.th (S.R.S.); winitacmu@gmail.com (W.P.); 4Department of Pharmaceutical Sciences, Faculty of Pharmacy, Chiang Mai University, Chiang Mai 50200, Thailand; 5Plant Bioactive Compound Laboratory (BAC), Department of Plant and Soil Sciences, Faculty of Agriculture, Chiang Mai University, Chiang Mai 50200, Thailand; 6Department of Chemistry, Faculty of Science, Chiang Mai University, Chiang Mai 50200, Thailand; 7Department of Biotechnology, Faculty of Technology, Khon Kaen University, Khon Kaen 40002, Thailand; alissara@kku.ac.th; 8Research Group for Development of Microbial Hydrogen Production Process, Khon Kaen University, Khon Kaen 40002, Thailand; 9Academy of Science, Royal Society of Thailand, Bangkok 10300, Thailand; 10Department of Chemical Technology and Environment, The University of Danang—University of Technology and Education, Danang 550000, Vietnam; ntmphuong@ute.udn.vn

**Keywords:** bacterial cellulose, biopolymer, carboxymethyl cellulose, CMC, nata de coco, sodium hydroxide

## Abstract

Bacterial cellulose from nata de coco was prepared from the fermentation of coconut juice with *Acetobacter xylinum* for 10 days at room temperature under sterile conditions. Carboxymethyl cellulose (CMC) was transformed from the bacterial cellulose from the nata de coco by carboxymethylation using different concentrations of sodium hydroxide (NaOH) and monochloroacetic acid (MCA) in an isopropyl (IPA) medium. The effects of various NaOH concentrations on the degree of substitution (DS), chemical structure, viscosity, color, crystallinity, morphology and the thermal properties of carboxymethyl bacterial cellulose powder from nata de coco (CMCn) were evaluated. In the carboxymethylation process, the optimal condition resulted from NaOH amount of 30 g/100 mL, as this provided the highest DS value (0.92). The crystallinity of CMCn declined after synthesis but seemed to be the same in each condition. The mechanical properties (tensile strength and percentage of elongation at break), water vapor permeability (WVP) and morphology of CMCn films obtained from CMCn synthesis using different NaOH concentrations were investigated. The tensile strength of CMCn film synthesized with a NaOH concentration of 30 g/100 mL increased, however it declined when the amount of NaOH concentration was too high. This result correlated with the DS value. The highest percent elongation at break was obtained from CMCn films synthesized with 50 g/100 mL NaOH, whereas the elongation at break decreased when NaOH concentration increased to 60 g/100 mL.

## 1. Introduction

Primary cell walls of eukaryotic plants, algae and the oomycetes consist of cellulose as the major component. The cellulose consists of D-glucose units linked as a linear chain, ranging from several hundred to over ten thousand β (1→4) units [1]. Cellulose is insoluble in water. However, sodium carboxymethyl cellulose (CMC), which is one of cellulose’s derivatives, can be dissolved in water [2]. CMC is an anionic, linear polymer, water-soluble cellulose reacted with monochloroacetic acid (MCA) or monochloroacetate acid (NaMCA). CMC is an important cellulose derivative applied in several industrial fields, such as the food industry, cosmetics, pharmaceuticals, detergents, textiles, [3] ceramics [4], etc. However, intensive utilization of wood has caused environmental problems and is expensive to manufacture. Therefore, there have been many studies about utilizing agricultural waste to be sources of CMC, such as cellulose from papaya peel [4], sugar beet pulp [5], sago waste [6], mulberry paper [7], *Mimosa pigra* peel [8] and durian husks [2,9], palm bunch and bagasse [10] and asparagus stalk ends [11]. The uses of CMC in food manufacturing require high purity of CMC grades ranging between 0.4 and 1.5 [12].

Cellulose has been obtained from bacterial cellulose (BC), including the genera *Agrobacterium*, *Rhizobium*, *Pseudomonas*, *Sarcina* and *Acetobacter* [13]. Acetobacter produced a pure bacterial cellulose aggregate containing impurities, such as hemicelluloses, pectin and lignin [14,15]. There are many desirable properties of bacterial cellulose, such as high purity, high degree of polymerization, high crystallinity, high wet tensile strength, high water-holding capacity and good biocompatibility [16,17]. *Acetobacter xylinum* is an acetic acid bacterium which can ferment and digest the carbon source in the coconut juice before converting it to the extracellular polysaccharide or cellulose [12]. The coconut juice is a waste product from the coconut milk industry known for its use in manufacturing cellulose [15,18]. Bacterial cellulose becomes a cellulosic white-to-creamy-yellow product called nata de coco. Therefore, the high intrinsic purity of bacterial cellulose from nata de coco together with the low environmental impact of the bacterium isolation means it is used in several applications, such as hydrogels [19], tissue engineering [20], cell culture [21], wound dressing and cancer therapy [22], CMC production [12], etc. However, the study of production and characterization of CMC films from nata de coco (CMCn) is limited.

This study aims to determine the effect of NaOH concentrations (20–60 g/100 mL) on the thermal properties, degree of substitution (DS), chemical structure, viscosity, crystallinity, and morphology of CMCn powder. With this aim, the effects of NaOH concentrations on solubility, mechanical properties (tensile strength, percentage elongation at break), percentage of transmittance, water vapor permeability (WVP) and the morphology of CMCn films were evaluated.

## 2. Materials and Methods

### 2.1. Materials

Coconut juice was collected from Muang District, Chiang Mai Province, Thailand. *Acetobacter xylinum* came from the Division of Biotechnology, Faculty of Agro-Industry, Chiang Mai University, Thailand. Di-ammonium hydrogen orthophosphate was obtained from Univar, Williamson Rd Ingleburn, Australia; magnesium sulphate from Prolabo, England; and commercial grade citric acid and sugar powder from the Thai Roong Ruang Sugar Group, Thailand. Glacial acetic acid and sodium hydroxide (artificial grade) were purchased from Lab-scan; hydrochloric acid and sodium chloride were purchased from Merck, Darmstadt, Germany; m-cresol purple, indicator grade, from Himedia, Marg, Mumbai, India; chloroacetic acid from Sigma-Aldrich, Burlington, MA, USA; isopropyl alcohol (IPA), ethanol and absolute methanol were purchased from Union Science, Muang District, Chiang Mai, Thailand.

### 2.2. Preparation of Bacterial Cellulose

Twenty liters of coconut juice were boiled in a pot and then 20 g of di-ammonium hydrogen orthophosphate, 10 g of magnesium sulphate, 30 g of citric acid and 100 g of sugar powder were added and mixed in. One and a half liters of the mixture were poured into each sterile plastic tray and then 55 mL of 95% *v*/*v* of ethanol was added. The mixture stood at room temperature for 30 min. One hundred milliliters of *Acetobacter xylinum* were then added to each tray with an aseptic technique. The tray was covered with fabric and paper to protect it from any contaminants. Nata de coco was obtained after the mixture had stood at room temperature for 10 days without being disturbed. The nata de coco was sliced, boiled 5 times in a pot with 15 L of water and dried in a hot air oven (Model: 1370FX-2E, Sheldon Manufacturing Inc., Cornelius, OR, USA) at 55 °C for 12 h. It was then ground using a grinder (Philips-HR1701, Simatupang, Jakarta, Indonesia), and screened through a 0.5 mm mesh sieve (35 mesh size). The nata de coco bacterial cellulose powders were contained in propylene bags until used.

### 2.3. Synthesis of Carboxymethyl Cellulose from Bacterial Cellulose 

Fifteen grams of bacterial cellulose powder from nata de coco, NaOH solution (100 mL) in different concentrations (30, 40, 50 and 60 g/100 mL) and 450 mL of isopropanol (IPA) were mixed for 30 min. Eighteen grams of monochloroacetic acid (MCA) was added to initiate the carboxymethylation process and the mixture was stirred continuously at 55 °C for 30 min. The mixture beaker was enclosed by aluminum foil and stood in a hot air oven at 55 °C for 3.5 h. After the heating process, the mixture separated into solid and liquid phases. The solid phase was collected and suspended in 300 mL of absolute methanol, and then 80 mL/100 mL of acetic acid was added to neutralize it. The suspended mixture was filtered by a Buchner funnel. Undesirable by-products were removed from the final product by soaking 5 times in 300 mL of 70% *v*/*v* ethanol for 10 min. Then, the final product was washed again with 300 mL of absolute methanol. The carboxymethylated bacterial cellulose (CMCn) obtained from nata de coco was dried in an oven at 55 °C for around 12 h [8].

### 2.4. Degree of Substitution (DS)

The degree of substitution (*DS*) of CMCn is defined as the average number of hydroxyl groups in the cellulose structure which have been replaced with carboxymethyl and sodium carboxymethyl groups at C2, 3 and 6. The DS determination was described in a crosscarmellose sodium monograph in USP XXIII. The method included two steps: titration and residue on ignition [8]. The DS is calculated using the following Equation (1):(1)DS= A + S
where *A* is the degree of substitution with carboxymethyl acid and *S* is the degree of substitution with sodium carboxymethyl. *A* and *S* were calculated using information from the titration and ignition steps using the following Equations (2) and (3):(2)$$A=\frac{1150M}{\left(7120-412M-80C\right)}$$
(3)$$S=\frac{\left(162+58A\right)C}{\left(7102-80C\right)}$$
where *M* is the *mEq* of the base required for endpoint titration. *C* is the percentage of ash remaining after ignition. The reported *DS* values are means of three repetitions.

### 2.5. Titration

Precisely one gram of CMCn was weighed and added to a 500 mL Erlenmeyer flask. Three hundred milliliters of sodium chloride (NaCl) solution (10 g/100 mL) was added next. The Erlenmeyer flask was covered by a stopper and intermittently shaken for 5 min. The mixture was mixed with 5 drops of m-cresol purple, and then 15 mL of 0.1 N hydrochloric acid (HCl) was added. If the solution color did not change, 0.1 N HCl was gradually added until the solution changed to yellow. The solution was gradually titrated with 0.1 N NaOH. The solution changed from yellow to violet at the endpoint. The net amount of milliequivalent base required for neutralization of 1 g of CMCn (M) was calculated using Equation (4):(4)$$M\left(mEq\right)=mmole\times Valence$$
where *m* is 10^−3^, mole is mass in grams per molecular weight of NaOH and the valence of NaOH is 1.

### 2.6. Residue on Ignition

A crucible was placed in an oven at 100 °C for 1 h and kept in a desiccator until the weight achieved a constant value (weighing apparatus, AR3130, Ohaus Corp. Pine, Brook, NJ, USA). Next, 1.000 g of CMCn was added to a crucible. To obtain black residue, a crucible containing CMCn was ignited in a 400 °C kiln (Carbolite, CWF1100, Scientific Promotion, Parsons Ln, Hope S33 6RB, England) for 1–1.5 h and then placed into a desiccator. The whole residue was wetted by adequate sulfuric acid and then heated until white fumes completely volatilized. White residue was obtained from the ignition crucible containing residue at 800 ± 25 °C and the residue was placed in desiccator to reach an accurate weight. All experiments were performed three times. The percentage of residue on ignition was calculated by following Equation (5).
(5)$$C=\frac{Weight\;of\;residue}{Weight\;of\;CMC}\times 100$$

### 2.7. Fourier Transform Infrared Spectroscopy (FTIR)

An infrared spectrophotometer (Bruker, Tensor 27, Fahrenheitstr 4, D-28359, Bremen, Germany) was used to evaluate the functional groups of bacterial cellulose from nata de coco and CMCn. The bacterial cellulose from nata de coco and CMCn samples (~2 mg) with KBr were used to make pellets. Transmission was measured at a wave number range of 4000–400 cm^−1^.

### 2.8. X-Ray Diffraction (XRD)

X-ray diffraction patterns of bacterial cellulose from nata de coco and CMCn were recorded in a reflection mode on a JEOL JDX-80-30 X-ray diffractometer, Brucker, Milton, ON, Canada. The scattering angle (2Ө) was from 5° to 60° at a scan rate of 2°/min.

### 2.9. Viscosity

The bacterial cellulose from nata de coco and CMCn were analyzed by a Rapid Visco Analyzer (Model: RVA-4, Newport Scientific, Warriewood, Australia) to measure viscosity. Three grams of cellulose and CMCn was dissolved in 25 mL of distilled water, heated to 80 °C and continuously stirred for 10 min. The sample solutions were tested in 2 steps. In the first step, the samples were tested at 960 rpm for 10 s. Next, the temperature was set at 30, 40 and 50° C at 5 min-intervals with a speed of 160 rpm. All tests were repeated three times [8]. 

### 2.10. Scanning Electron Microscopy (SEM)

The morphology of bacterial cellulose from nata de coco, CMCn powder and CMCn films were analyzed using a scanning electron microscope (SEM) (Phillip XL 30 ESEM, FEI Company, Hillsboro, OR, USA) equipped with a large field detector. The acceleration voltage was 15 kV under low settings with 5000×. 

### 2.11. Thermal Properties

The thermal properties of the melting point of bacterial cellulose from nata de coco (CMCn) were determined using a differential scanning calorimeter (DSC) (Perkin Elmer precisely, Inst. Model Pyris Diamond DSC, Hodogaya-Ku, Godocho, Kanagawa, Japan). Aluminum pans containing 10 mg of samples were heated from 40 to 240 °C. A heat rate was set as 10 °C min^−1^. The tests were performed under N_2_ gas with a flow rate of 50 mL min^−1^. The data was represented in terms of a thermogram, which indicated the melting point. All samples were tested three times [8]. 

### 2.12. Film Preparation

To obtain a film-forming solution, 3 g (Weighing apparatus, AR3130, Ohaus Corp. Pine, Brook, NJ, USA) of CMCn was dissolved in 300 mL of 80 °C distilled water with constant stirring for 15 min. Glycerol (15 g/100 g) was added and the film-forming solution was casted onto plates (20 cm × 15 cm). The plates containing the film-forming solution was placed at room temperature for 48 h to obtain dry CMCn films. Then, the CMCn films were removed from the plates. The CMCn films were kept at 27 ± 2 °C with 65 ± 2% relative humidity (RH) for 24 h (Thai industrial standard, TIS 949-1990).

### 2.13. Film Solubility

The method to determine the percentage of solubility of CMCn films was modified from Phan et al. [23]. CMCn films were dried at 105 °C in a hot air oven for 24 h and kept in desiccators for 24 h. Then, about 0.2000 g of the CMCn films was weighed. This weight was recorded as an initial dry weight (*W_i_*). Each weighed CMCn film was suspended in 50 mL of distilled water with shaking at 500 rpm for 15 min and poured onto filter paper (Whatman, No.93). The films then were dried in an oven at 105 °C for 24 h and weighed to obtain the final dry weight (*W_f_*). The percentage of soluble matter (%SM) of the films was calculated by using the following Equation (6):(6)$$\%\mathrm{SM}=\frac{\left(W_{i}-W_{f}\right)}{W_{i}}$$

### 2.14. Percent Transmittance

The percent transmittance of CMCn film was measured using a spectrophotometer (LaboMed, inc., Los Angeles, CA, USA) with a wavelength of 660 nm.

### 2.15. Mechanical Properties

The CMCn films were cut into rectangular strips (1.5 cm × 14 cm) for tensile strength and percentage elongation measurements. To precondition them, CMCn films were kept for 24 h at 27 ± 2 °C and relative humidity (RH) was controlled in the range of 65 ± 2% (Thai Industrial Standard; TIS 949-1990). Film thickness was measured at five different location in each sample by a micrometer, model GT-313-A (Gotech testing machine Inc., Taichung Industry Park, Taichung City, Taiwan). Tensile strength and water vapor permeability (WVP) was calculated using average film thickness. The tensile strength (TS) and percentage elongation at break (EB) were measured using a Universal Testing Machine Model 1000 (HIKS, Selfords, Redhill, England) as per the ASTM Method (ASTM, D882-80a, 1995a). The upper grips were separated. The machine was operated at 100 mm and 20 mm min^−1^ for the initial grip separation distance and crosshead speed, respectively. To calculate the TS, the maximum load at break was divided by the cross-sectional area of the sample. The EB was defined by the increasing of sample length at break point as the percentage of the initial length (100 mm). All mechanical tests were repeated 10 times.

### 2.16. Water Vapor Permeability (WVP)

The method for measuring the water vapor permeability (WVP) of the CMCn films was described in the ASTM method (ASTM, E96-93, 1993). Circular aluminum cups (8 cm diameter and 2 m depth) containing 10 g of silica gel were covered by the circle shape of CMCn films (7 cm diameter). Then, the aluminum cups were closed by paraffin wax, weighed and kept in a desiccator. NaCl saturated solution was used to maintain the conditions (25 °C, 75% RH) in the desiccator. The closed cups were weighed daily for 10 days. A slope was obtained from plotting a graph between weight gain (Y axis) and time (X axis). The water vapor transmission rate (WVTR) was calculated using the following Equation (7):(7)$$WVTR\;\left(\frac{g}{m^{2}}.day\right)=\;\frac{slope}{Film\;Area}$$
where the film area was 28.27 cm^2^ and the WVP (g.m/m^2^.mmHg.day) was calculated using the following Equation (8):(8)$$WVP\;\left(g.\frac{m}{m^{2}}.day.mmHg\right)=\frac{WVTR}{∆P}\times L$$
where *L* is the average film thickness (mm) and ∆*P* is the partial water vapor pressure difference (mmHg) across two sides of the film specimen (the vapor pressure of pure water at 23.6 °C = 21.845 mmHg). These samples were tested 3 times [8]. 

### 2.17. Statistical Analysis

All the data were presented as the mean ± SD. One-way ANOVA was used to evaluate the significance of differences at the significance level of *p*-value < 0.05. Statistical analysis was performed using SPSS software version 16.0 (SPSS Inc., Chicago, IL, USA).

## 3. Results and Discussion

### 3.1. Degree of Substitution (DS)

The effects of NaOH concentrations on the DS of CMCn is shown in Figure 1. As NaOH concentrations increased from 20 to 30 g/100 mL NaOH, the DS of CMCn increased as well, while the DS dropped at 40 to 60 g/100 mL NaOH concentration. The values of the obtained CMCn were from 0.30–0.92. This phenomenon can be explained by the carboxymethylation procedure, where two competitive reactions occurred concurrently. A cellulose hydroxyl reacts with sodium monochloroacetate (NaMCA) to obtain CMCn in the first reaction, as shown in Equations (9) and (10).
CLL–OH + NaOH →CLL–ONa + H_2_O(9)
(Cellulose)   (Reactive alkaline form)
CLL–ONa + Cl–CH_2_–COONa → CLL–O–CH_2_–COONa + NaCl(10)
     (CMC)

NaMCA alternated to sodium glycolate as a byproduct by reacting with NaOH in the second reaction, as shown in Equation (11).
NaOH + Cl–CH_2_COONa → HO–CH_2_COONa + NaCl(11)

The second reaction overcame the first at stronger alkaline concentrations. In conditions of excessive alkalinity, DS was low because a side reaction dominates, causing the formation of sodium glycolate as a byproduct. This result agrees with Rachtanapun and Rattanapanone [8], who reported that degradation of CMC from *Mimosa pigra* occurred because of high concentrations of NaOH. Similar results were reported for the maximum DS value (0.87) of carboxymethyl cellulose from durian rind [2] and the maximum DS value (0.98) of carboxymethyl cellulose from asparagus stalk ends with a NaOH concentration of 30 g/100 mL [11].

### 3.2. Percent Yield of Carboxymethyl Cellulose from Nata de Coco

The effect of NaOH concentrations on percent yield of carboxymethyl bacterial cellulose from nata de coco (CMCn) powder synthesized is shown in Figure 2. This study investigated the effect of NaOH in concentrations of 20–60 g/100 mL on the percent yield of bacterial cellulose powder synthesis. At 20–40 g/100 mL NaOH concentrations, the percentage yield increased. However, the percentage decreased at 50–60 g/100 mL NaOH concentrations. This resulted from the alkali-catalyzed degradation of bacterial cellulose. The trend of percentage yield of CMCn was similar to the DS results (Figure 2). This result was in agreement with carboxymethyl cellulose from durian husks [2] and asparagus stalk ends [11]. 

### 3.3. Fourier Transform Infrared Spectroscopy (FTIR)

The substitution reaction in carboxymethylation was identified using FTIR. The FTIR spectrums of bacterial cellulose from nata de coco and CMCn synthesized with a 30 g/100 mL NaOH concentration are shown in Figure 3. The same functional groups appeared in both cellulose and CMCn. A broad peak at 3441 cm^−1^ referred to the hydroxyl group (–OH stretching). A peak at 1420 cm^−1^ was related to –CH_2_ scissoring. Strong peaks at 1604 cm^−1^ and 1060 cm^−1^ indicated carbonyl group (C=O stretching) and ether groups (–O– stretching), respectively [8]. In the CMCn sample, the obvious increase of the carbonyl group (C=O), methyl group (–CH_2_) and ether group (–O–) peaks was observed, but the intensity of the hydroxyl group (–OH) was lower when compared to bacterial cellulose (Figure 3). This result indicated that cellulose molecules changed due to carboxymethylation occurrence [2,6,10,11,17].

### 3.4. Effect of Various NaOH Concentrations on Viscosity of CMCn

The effect of using various NaOH concentrations on the viscosity of CMC is shown in Figure 4. The peak viscosity of CMCn increased when the concentration of NaOH increased from 20 to 30 g/100 mL because hydroxyl groups on the cellulose molecule were substituted by more carboxymethyl groups, being hydrophilic groups. Cohesive forces reduced because CMCn temperature increased, whereas the rate of molecular interchange concurrently increased [8,24]. The viscosity of CMCn solution was in accordance with the DS value at the same temperature. A previous study found that NaOH concentrations influenced the viscosity of CMC [25]. They stated that more carboxymethyl groups (a higher DS value) increased CMC viscosity [8]. In addition, they reported that the degradation of CMC polymers causes decreasing viscosity with too much NaOH, leading to a lower DS [26].

### 3.5. Effects of Various NaOH Concentrations on Thermal Properties of CMCn Powder

The effects of various NaOH concentrations on the thermal properties of bacterial cellulose from nata de coco and CMCn powder are shown in Figure 5. The melting temperature ™ of bacterial cellulose was 167.4 °C, and those of CMCn synthesized with 20, 30, 40, 50 and 60 g/100 mL NaOH were 175.3, 188.2, 179.4, 164.7 and 151.0 °C, respectively. The melting temperature of CMCn was increased at 20–30 g/100 mL because the substituent of carboxymethyl groups caused increases in the ionic character and intermolecular forces between the polymer chains. At 40–60 g/100 mL NaOH, the melting temperature decreased because of the side reaction predominating with the formation of sodium glycolate as a byproduct and chain breaking of the CMCn polymer. This result was in agreement with carboxymethyl cellulose from asparagus stalk ends [11].

### 3.6. X-Ray Diffraction (XRD)

The effect of NaOH concentrations on the amount of crystallinity in bacterial cellulose from nata de coco and CMCn powder is shown in Figure 6. The treatment of cellulose with NaOH caused a decrease in the amount of cellulose crystallinity because NaOH leaded to the dividing of hydrogen bond [18]. By comparison with cellulose without alkalization with NaOH, monochloroacetic acid molecules substituted into bacterial cellulose molecules more easily because the distance between each polymer molecule increased. Thus, before the carboxymethylation reaction, NaOH had an effect on cellulose structured to decrease the crystallinity of the CMCn. These results were also found in the reduction of crystallinity of CMC from asparagus stalk ends [11] and cavendish banana cellulose [26] due to alkalizing by 20 g/100 mL and 15 g/100 mL, respectively. 

### 3.7. Scanning Electron Microscopy of Cellulose from Nata de Coco and CMCn Powder

The scanning electron micrographs of bacterial cellulose from nata de coco and CMCn powder with 20, 30, 40, 50 and 60 g/100 mL NaOH are shown in Figure 7a–f. The micrograph of bacterial cellulose from nata de coco (Figure 7a) exhibited a compact appearance with a smooth surface without pores or cracks, and a lot of small fibers appeared. However, the microstructure of samples dramatically changed when the NaOH concentration increased. The morphology of CMCn powder with 20 g/100 mL NaOH (Figure 7b) showed small fibers with minimal damage. As shown in Figure 7c, the morphology of CMCn powder with 30 g/100 mL NaOH was compact and dense, with no signs of cracks and pits. The morphology of CMCn powder with 40 g/100 mL NaOH (Figure 7d) was deformed and had some cracks and pits. Moreover, the morphology of CMCn powder with 40 g/100 mL NaOH also correlated with the DS value, due to a chain degradation of the CMC polymer. This result is similar to those of other studies [2,8,11]. CMCn powder with 50 g/100 mL NaOH (Figure 7e) showed increased surface irregularities with a more indented and collapsed surface. Moreover, when increasing NaOH concentration to 60 g/100 mL (Figure 7d), the surface became rougher and totally deformed. Thus, it can be concluded that synthesis with increasing NaOH concentrations damages the surface area of bacterial cellulose powder. This result was similar to those of carboxymethyl rice starch [27] and carboxymethyl cassava starch [28] and carboxymethyl cellulose from asparagus stalk ends [11]. The principal parameter for weakening the structure of cellulose and causing a loss of crystallinity that allowed etherifying agents to reach the cellulose molecules for carboxymethylation processes was the alkalization [27]. Consequently, this result was consistent with the DS value. To indicate carboxymethyl reactions, the color formation was investigated by color measurement. The main effects on CMC color value were increasing a* (redness), b* (yellowness) and yellowness index (YI) values as NaOH concentrations increased (20–30 g/100 mL NaOH). The L* value of CMCn synthesized with various NaOH concentrations decreased as the concentrations increased from 20 to 30 g/100 mL NaOH. The a* and b* values decreased, but the L* value of CMCn increased as NaOH was increased to 40 g/100 mL NaOH. Moreover, the trend of YI of CMCn was decreased when increasing the NaOH concentration up to 40%. This phenomenon was probably due to the first step in the carboxymethylation of cellulose (Equation (9)), which delivered CMC or sodium glycolate [10]. At a high NaOH concentration (50 and 60 g/100 mL NaOH), all color values were decreased. A carboxymethylation reaction might have been a reason for the color change in this study [2,11,27]. The ∆E of cellulose and CMCn had same trends as the a* and b* values. The WI of cellulose and CMCn had the same trends as the L* value (Table 1).

### 3.8. Solubility and Transmittance

The effects of NaOH concentrations on the solubility of CMCn films are shown in Figure 8. At 20–30 g/100 mL NaOH, the solubility of CMCn film increased; however, at 40–60 g/100 mL NaOH, the film solubility decreased. The percentage of soluble matter could be indicated by the DS. The percentage of soluble matter rose as the DS value increased [6]. The effects of NaOH concentrations on the percent transmittance of the CMCn films are shown in Figure 9. The percent transmittance increased with NaOH concentrations of 20–30 g/100 mL and decreased at 40–60 g/100 mL. This result correlated with the percentage of solubility of the CMCn films.

### 3.9. Scanning Electron Microscopy of Cellulose from Nata de Coco and CMCn Films

Cross-section views of scanning electron micrographs of CMCn films synthesized with various NaOH concentrations are shown in Figure 10a–e. As shown in Figure 10a, the morphology of CMCn films with 20 g/100 mL NaOH was a rather smooth surface without pits and cracks. CMCn films with 30 g/100 mL NaOH (Figure 10b) were compact and dense. The morphology of CMCn films synthesized with 40 g/100 mL NaOH (Figure 10c) was still compact but rougher and some scraps protruded on the surface. CMCn films with 50 g/100 mL NaOH (Figure 10d) showed increasing surface irregularities and defects such as cracks and pits. The morphology of CMCn films with 60 g/100 mL NaOH (Figure 10e) was deformed and revealed lots of cracks and some scraps.

### 3.10. Water Vapor Permeability (WVP)

The effects of NaOH concentrations on the water vapor permeability (WVP) of the CMCn films are shown in Figure 11. When the concentration of NaOH increased, cellulose transformed to CMCn, causing a higher polarity. This result led to a rise of the WVP. Moreover, the morphology of CMCn films with 40–60 NaOH showed cracks and pits. Damage dramatically increased in the WVP of CMCn with 50–60 g/100 mL NaOH. This result can be explained through a SEM micrograph (Figure 10). SEM morphology demonstrated that CMCn films with 40–60 NaOH showed cracks and pits. The results were in agreement with studies of CMC from rice starch [24], which showed that the WVP of carboxymethyl rice starch films rose due to increases in NaOH concentrations. Furthermore, carboxylation influenced increases of polarity, reductions in crystallinity and changes in granule morphology [29].

### 3.11. Tensile Strength (TS) and Elongation at Break (%)

The tensile strength values of CMC films with various NaOH concentrations are shown in Figure 12a. When NaOH increased (20–30 g/100 mL NaOH), the tensile strength also rose. This is because TS values are correlated with increases in DS values because of the carboxymethyl group substitution, causing a rise in the ionic character and intermolecular forces between the polymer chains [23]. Moreover, this is related to why the morphology of CMCn film synthesized with 30 g/100 mL NaOH was compact and dense. However, at higher NaOH concentration, the TS dropped because of sodium glycolate, which is a secondary product from the CMC synthesis reaction and polymer degradation. These results are related to those of CMC films from *Mimosa pigra* [8], which shows that the TS of CMC films from *Mimosa pigra* rose when concentrations of NaOH increased (20–30 g/100 mL). Studies of CMC films from mulberry paper waste showed that the TS increased with increasing NaOH concentrations, but too-high NaOH concentrations (60 g/100 mL) could cause a hydrolysis reaction in the cellulose chain [7]. The percentage of elongation at break (EB) of CMCn films with various NaOH concentrations used in CMCn synthesis are shown in Figure 12b. CMCn films exhibited higher EB when NaOH concentrations increased (20–50 g/100 mL NaOH). However, when NaOH concentration was increased to 60 g/100 mL, the EB of CMCn films decreased. This phenomenon can be explained as a cellulose structure with decreased crystallinity caused the CMCn films to have increased elasticity due to higher concentrations of NaOH. Nevertheless, at 60 g/100 mL NaOH, the occurrence of cellulose hydrolysis resulted in lower flexibility of CMCn films [8]. 

## 4. Conclusions

From the results, this study was successful in using nata de coco as a resource to obtain carboxymethyl bacterial cellulose (CMCn) using different NaOH concentrations. This demonstrates that the major parameter relating to the properties of CMCn is NaOH quantity. The DS of CMCn dramatically increased when NaOH concentrations increased from 20 to 30 g/100 mL in CMCn synthesis, and the DS value gradually dropped at NaOH concentrations of 40–60 g/100 mL. The mechanical properties and viscosity of CMCn films were correlated with the DS. CMCn consisting of high DS can have improved mechanical properties. Therefore, the CMCn films synthesized with 30 g/100 mL NaOH exhibited good mechanical properties, such as high TS and EB with no evidence of cracking and pitting when examined by SEM. CMC is widely applied in several fields, including in the food, nonfood and pharmaceutical industries, and thus the results from this study are useful. The investigation of methods to modify cellulose in this study, such as varying the NaOH content, supports new and potentially useful applications of cellulose from nata de coco.

## Figures and Tables

**Figure 1 polymers-13-00348-f001:**
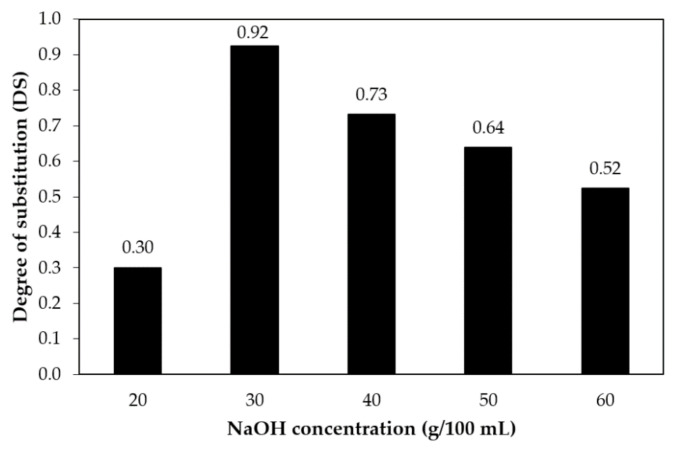
Effect of the amount of NaOH on the DS of CMCn. CMCn: carboxymethyl bacterial cellulose powder from nata de coco. DS: degree of substitution.

**Figure 2 polymers-13-00348-f002:**
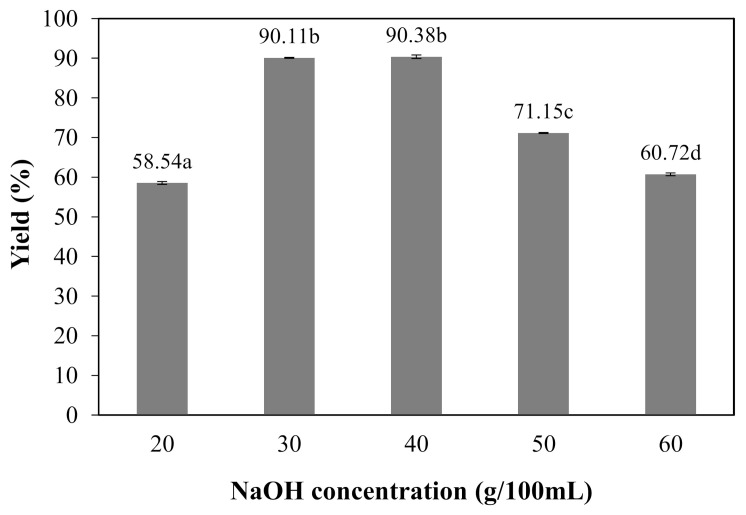
Percent yield of carboxymethyl cellulose from nata de coco synthesized with various NaOH concentrations (20, 30, 40, 50 and 60 g/100 mL). The different letter, e.g., ‘a’, ‘b’, ‘c’ or ‘d’ are statistically different (*p* < 0.05).

**Figure 3 polymers-13-00348-f003:**
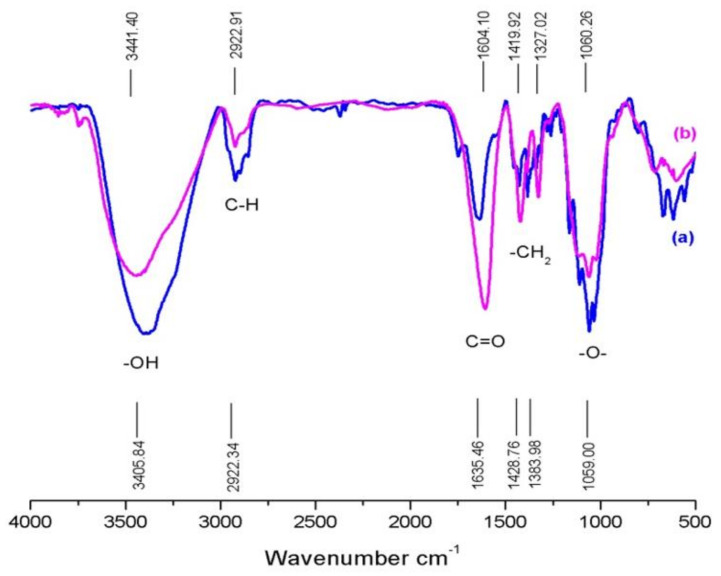
Fourier transform infrared spectroscopy of (**a**) bacterial cellulose from nata de coco and (**b**) CMCn synthesized with 30 g/100 mL NaOH.

**Figure 4 polymers-13-00348-f004:**
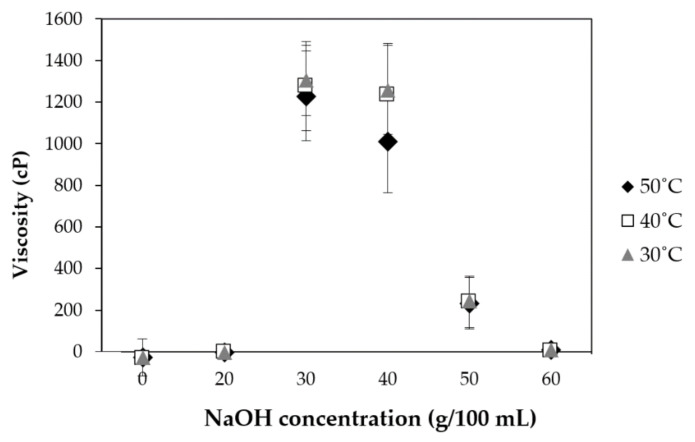
Effect of NaOH concentrations on the viscosity of bacterial cellulose and CMCn.

**Figure 5 polymers-13-00348-f005:**
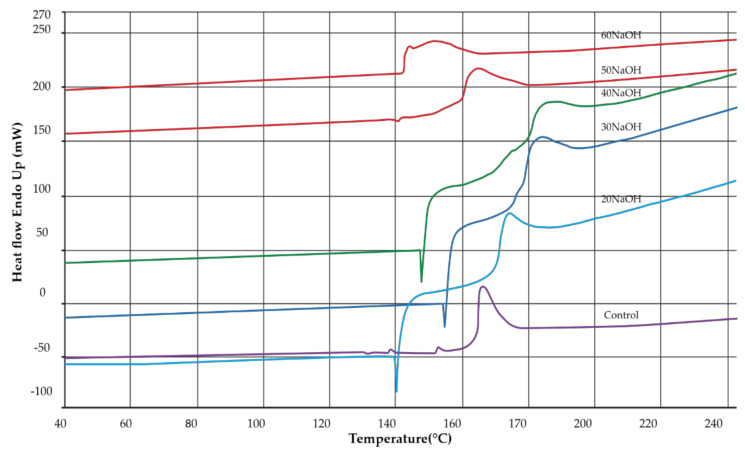
Differential scanning calorimetry of cellulose from nata de coco and CMCn synthesized with various amounts of NaOH.

**Figure 6 polymers-13-00348-f006:**
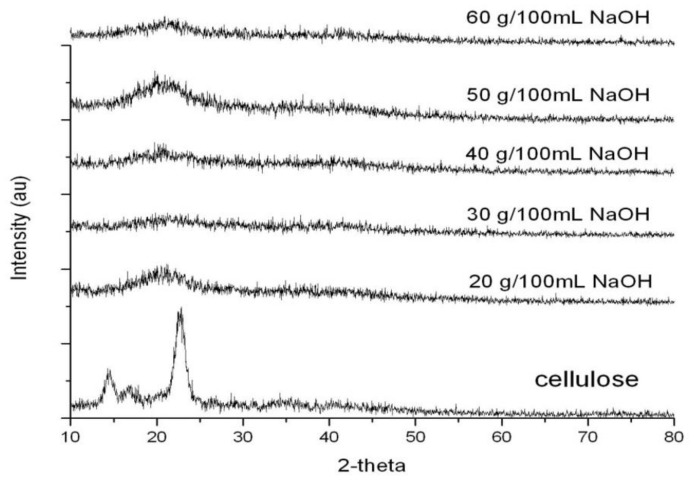
X-ray diffractograms of cellulose from nata de coco and CMCn synthesized with various amounts of NaOH.

**Figure 7 polymers-13-00348-f007:**
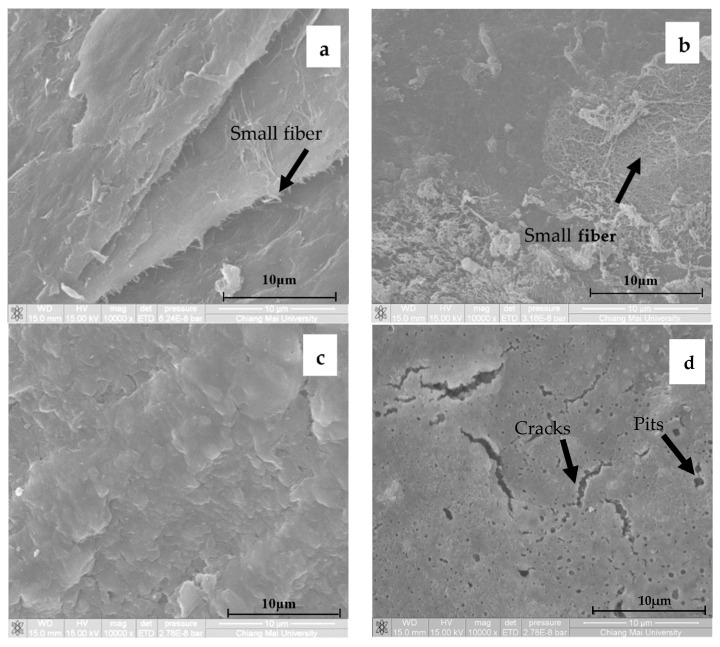
Scanning electron micrographs of (**a**) cellulose from nata de coco and CMCn powder: with (**b**) 20 g/100 mL NaOH, (**c**) 30 g/100 mL NaOH, (**d**) 40 g/100 mL NaOH, (**e**) 50 g/100 mL NaOH and (**f**) 60 g/100 mL NaOH.

**Figure 8 polymers-13-00348-f008:**
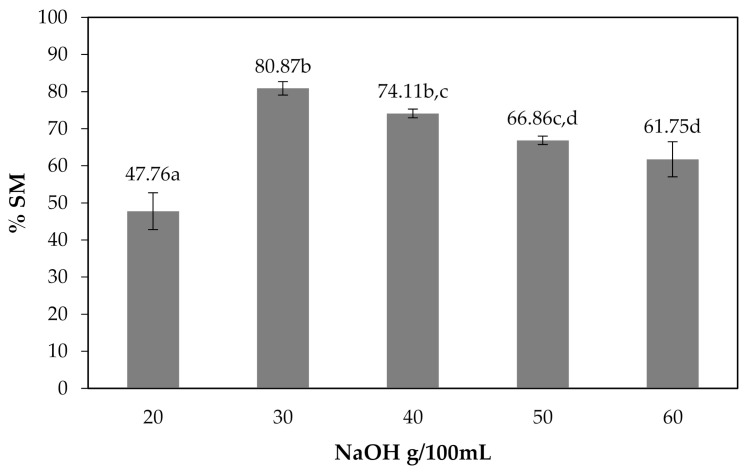
Effect of NaOH concentrations on the percentage of soluble matter of the CMCn films. The different letter, e.g., ‘a’, ‘b’, ‘c’ or ‘d’ are statistically different (*p* < 0.05).

**Figure 9 polymers-13-00348-f009:**
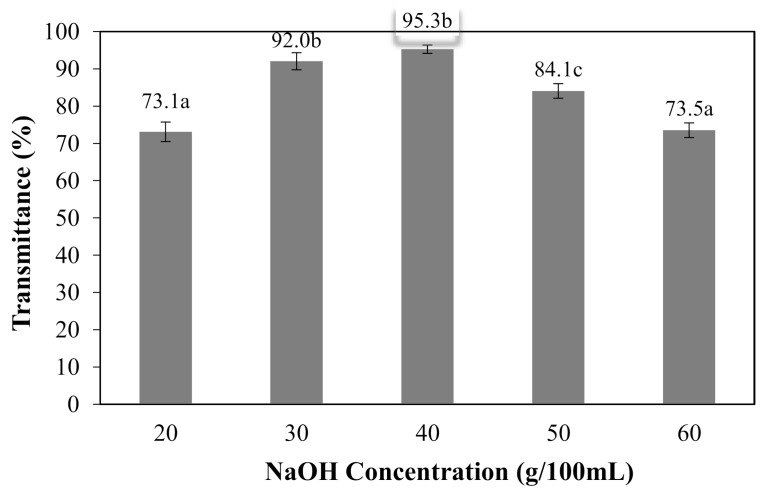
Effect of NaOH concentrations on the percentage of transmittance of the CMCn films. The different letter, e.g., ‘a’, ‘b’, ‘c’ or ‘d’ are statistically different (*p* < 0.05).

**Figure 10 polymers-13-00348-f010:**
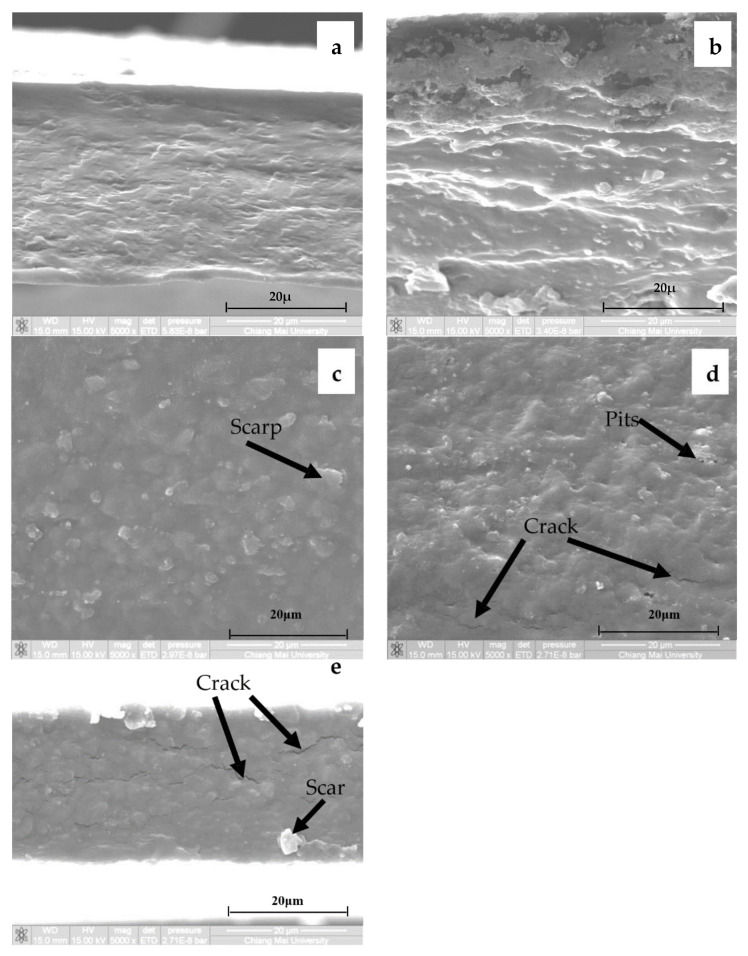
Scanning electron micrographs of CMCn films: with (**a**) 20 g/100 mL NaOH, (**b**) 30 g/100 mL NaOH, (**c**) 40 g/100 mL NaOH, (**d**) 50 g/100 mL NaOH and (**e**) 60 g/100 mL NaOH.

**Figure 11 polymers-13-00348-f011:**
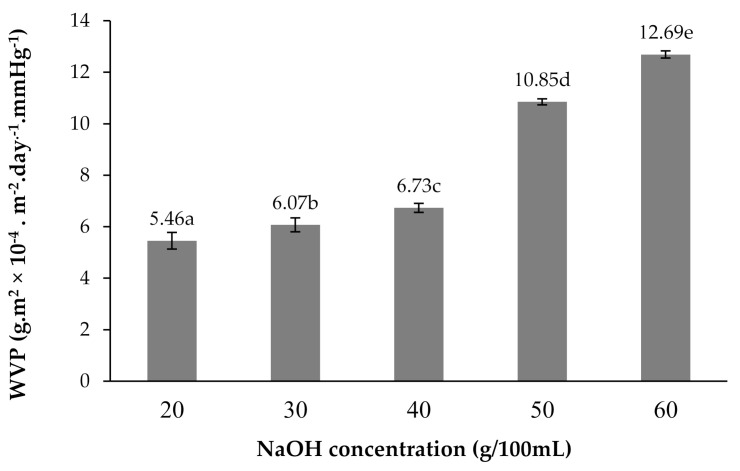
Effect of NaOH concentration (20, 30, 40, 50 and 60 g/100 mL) on the water vapor permeability (WVP) of CMCn films. The different letter, e.g., ‘a’, ‘b’, ‘c’ or ‘d’ are statistically different (*p* < 0.05).

**Figure 12 polymers-13-00348-f012:**
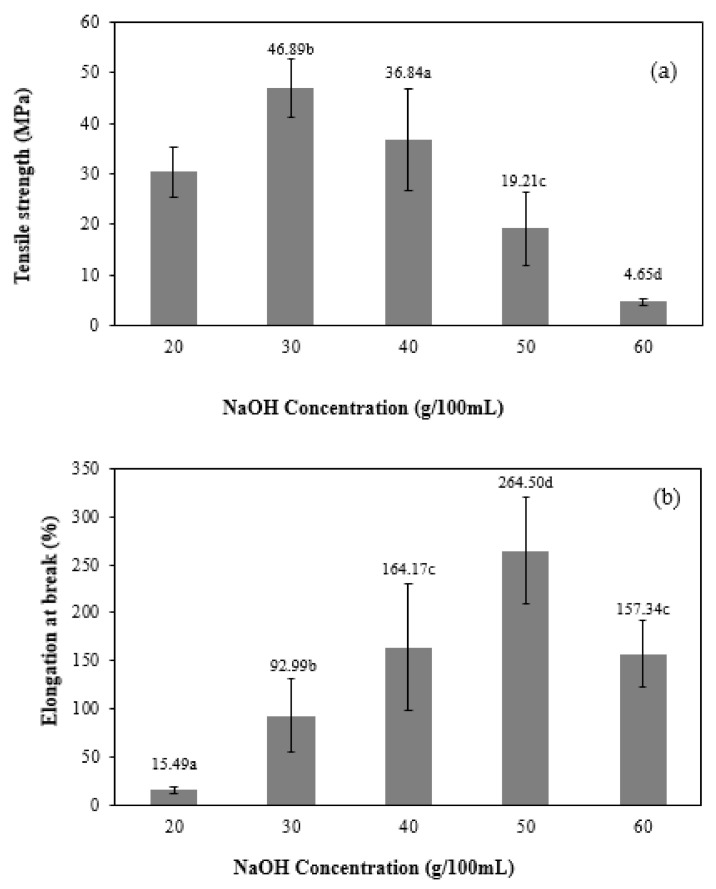
Effect of various NaOH concentrations (0, 20, 30, 40, 50 and 60 g/100 mL) on (**a**) tensile strength and (**b**) percent elongation at break of CMCn films. The different letter, e.g., ‘a’, ‘b’, ‘c’ or ‘d’ are statistically different (*p* < 0.05).

**Table 1 polymers-13-00348-t001:** Color values of bacterial cellulose from nata de coco and CMCn synthesized with various amounts of NaOH.

NaOH (g/100 mL)	L*	a*	b*	ΔE	YI	WI	h_ab_
0	75.62a	2.247a	16.62a	23.39a	31.42a	70.45a	82.34a
20	64.79b	8.23b	28.61b	40.29b	63.09b	54.70b	73.94b
30	65.54b	10.19c	28.96b	40.61b	63.26b	54.74b	70.60c
40	74.34a	7.23d	22.14c	29.08c	42.55c	65.99c	71.91d
50	70.01c	7.67e	21.64c	31.85d	44.22c	62.92d	70.49c
60	68.14d	7.47de	21.13c	32.89d	44.30c	61.67d	70.54c

Note: Obtained by Duncan’s test (*p* < 0.05). L* = lightness, a* = redness, b* = yellowness, ΔE = total color difference, YI = yellowness index, WI = whiteness index, h_ab_ = hue. The different letter, e.g., ‘a’, ‘b’, ‘c’ or ‘d’ are statistically different (*p* < 0.05).

## Data Availability

The data presented in this study are available on request from the corresponding author.

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
