# Peer review of "Carboxymethyl Bacterial Cellulose from Nata de Coco: Effects of NaOH"

_polymers, 2021, doi:10.3390/polym13030348_

Round 1

Reviewer 1 Report

Dear Authors,

the manuscript you prepared is describing the research of wide range of well-designed and completed tests. The results are properly evaluated, correctly shown and commented.

Below, please, find several comments I have to your work:

- line 103: is mesh size given in milimeters? Please add the proper units
- line 124 - equation (1) and remaining: where applicable, please add the units resulting from equations
- line 209: what was the precision of thickness measurement by micrometer? Also, what was the precision of weight measurement? Please add it in proper way.
- line 364 (table 1): how can you comment/explain such irregular results of “L” parameter?
- line 442, sentence: “The 30g/100mL NaOH-synthesized CMCn films exhibited the proper mechanical properties”. Since you did not refer your results to any document like standard, I’m afraid you are not allowed to comment you results as “proper mechanical properties”. In this way the word “proper” means nothing when not compared to the reference data.

With best regards!

Author Response

The manuscript you prepared is describing the research of wide range of well-designed and completed tests. The results are properly evaluated, correctly shown, and commented.

Thank you very much for this insightful comment about our manuscript.

Below, please, find several comments I have to your work:

  1. line 103: is mesh size given in milimeters? Please add the proper units

            A particular size of a sieve has been added in line 103.

  1. line 124 - equation (1) and remaining: where applicable, please add the units resulting from equations

Thank you for the comment from reviewer.

Degree of substitution (DS) value has no unit. DS is mean how many carboxymethyl can replace -OH group in cellulose. If DS is higher than 0.4, CMC will be soluble in water. But if DS is lower than 0.4, CMC will be not soluble in water.

  1. line 209: what was the precision of thickness measurement by micrometer? Also, what was the precision of weight measurement? Please add it in proper way.

A micrometer model GT-313-A has a unit in a range of 0-0.050 in. (0-1.27 mm) and designed for very thin materials such as thin plastic films, paper, non-woven textiles, board and battery separators which is high accuracy and repeatability. The precision of a scale is a measure of the repeatability of an object's displayed weight for multiple weightings of the same object. Therefore, all mechanical tests were repeated in 10 times which has already been added in line 218.

  1. line 364 (table 1): how can you comment/explain such irregular results of “L” parameter?

Thank you for your comment. The results and discussion of “L” parameter was added.

  1. line 442, sentence: “The 30g/100mL NaOH-synthesized CMCn films exhibited the proper mechanical properties”. Since you did not refer your results to any document like standard, I’m afraid you are not allowed to comment you results as “proper mechanical properties”. In this way the word “proper” means nothing when not compared to the reference data.

Thank you for your suggestion. We agree to change the word “proper” to “good” in line 442.

Reviewer 2 Report

The manuscript is interesting, but it has shortcomings that need to be corrected. It is a multi-author work, but the contribution of individual authors is not clearly specified in the part Author Contributions. 

In the part Results and Discussion, equation 9 shows that ROH alcohols react with NaOH, which is incompatible with the chemistry because only methanol reacts in this way, so it should be corrected because it is a factual error!

The conclusions should be developed and the research achievements emphasized more.

Author Response

The manuscript is interesting, but it has shortcomings that need to be corrected.

Thank you very much for this insightful comment about our paper.

  1. It is a multi-author work, but the contribution of individual authors is not clearly specified in the part Author Contributions.

            The contributions of the authors have been stated in lines 448 and 450.

  1. In the part Results and Discussion, equation 9 shows that ROH alcohols react with NaOH, which is incompatible with the chemistry because only methanol reacts in this way, so it should be corrected because it is a factual error!

Thank you for very good comment. I am sorry. I make you misunderstand.

I have already made it clear.

ROH in this reaction is not alcohol but it is cellulose.

ROH is cellulose. I change from “ROH” to “CLL-OH”.

Please see the Equation 9 with yellow highlight.

(Reactive alkaline form)

(Cellulose)

CLL–OH  +  NaOH  → CLL–ONa  +  H2O

(1)

(CMC)

CLL–ONa + Cl–CH2–COONa → CLL–O–CH2–COONa + NaCl

(2)

  1. The conclusions should be developed, and the research achievements emphasized more.

Conclusions was developed.

We believe we have attended to all the corrections suggested. We have also made small changes to our text to make some other points clearer. Thank you for taking the time and energy to help us improve the paper.

Sincerely,

Assoc. Prof. Pornchai Rachtanapun
